# Design and Implementation of a Ball-Plate Control System and Python Script for Educational Purposes in STEM Technologies

**DOI:** 10.3390/s22051875

**Published:** 2022-02-27

**Authors:** Vladimir Tudić, Damir Kralj, Josip Hoster, Tomislav Tropčić

**Affiliations:** Department of Mechanical Engineering, Karlovac University of Applied Sciences, 47000 Karlovac, Croatia; damir.kralj@vuka.hr (D.K.); josip.hoster@vuka.hr (J.H.); tomislav.tropcic@gmail.com (T.T.)

**Keywords:** Ball-Plate System, STEM, USB HD camera, Python scripts, ready-made functions, PID controller

## Abstract

This paper presents the process of designing, fabricating, assembling, programming and optimizing a prototype nonlinear mechatronic Ball-Plate System (BPS) as a laboratory platform for engineering education STEM. Due to the nonlinearity and complexity of BPS, the task presents challenges such as: (1) difficulty in controlling the stabilization of a particular position point, known as steady-state error, (2) position resolution, known as specific distance error, and (3) adverse environmental effects—light-shadow error, which is also discussed in this paper. The laboratory prototype BPS for education was designed, manufactured and installed at Karlovac University of Applied Sciences in the Department of Mechanical Engineering, Mechatronics program. The low-cost two-degree BPS uses a USB HD camera for computer vision as a feedback sensor and two DC servo motors as actuators. Due to control problems, an advanced block diagram of the control system is proposed and discussed. An open-source control system based on Python scripts, which allows the use of ready-made functions from the library, allows the color of the ball and the parameters of the PID controller to be changed, indirectly simplifying the control system and performing mathematical calculations directly. The authors will continue their research on this BPS mechatronic platform and control algorithms.

## 1. Introduction

Engineering students in STEM need the practical application of theoretical concepts learned in class to master the methods and problems of controlling. The author’s goal is to help students learn the control theories of systems in an engineering context through the design and implementation of a simple and low-cost BPS. Students will be able to apply computer modeling tools, control the system design and achieve software–hardware implementation in real-time while solving the ball position control problem. The overall project development is presented and can be adopted as a guide for replicating the results or as a basis for a new approach to the design of mechatronic learning platforms. In both cases, we have a tool for implementing and evaluating experimentally controlled strategies that can be further improved in the future. University laboratories and experiments play a very important role in successful education in STEM engineering, especially when it comes to robotics and automatic control applications. The rapid development of BPS applications was noted recently due to the challenges related to control and fast dynamic response, which requires short and fast sensing and immediate correction of the selected controller. Since control of fast unstable systems is very important in a variety of practical applications, a mechatronic learning platform BPS can be a successful tool when used for training in robotics and automation control applications and control methods. In the literature, we find several examples of approaches to this topic.

The feedback of the position of a sphere is detected with the help of a camera, as shown in [1]. The article describes the synthesis of a controller for a two-dimensional electromechanical system consisting of the ball and a plate, intended for a study of system dynamics and laboratory experiments with various control methods based on classical and modern control theory. The system consists of a square plate movably fixed in the center. Its inclination can be changed in two orthogonal directions. A servo drive with a controller and two stepper motors was used to tilt the plate. The control problem of the described system is to keep the freely rolling ball in a certain position on the plate. An intelligent video system consisting of a CCD camera, an image interface and a program for real-time image processing is used to measure the position of the ball.

The BPS was also understood as the two-dimensional movement of the sphere and beam system presented in [2]. The author S. Awtar and others presented the dynamic properties of the BPS, the mathematical model with the corresponding simplified model and the analysis of the applications of different types of PID controllers. Based on the results of the analysis of different controllers, a controller with a switching mechanism is proposed to control the position of the BPS [3]. In addition, F. Zheng describes in [4] the design of the hardware, the selection of sensors and actuators, the modeling of the system, the identification of the parameters, the design of the controller and experimental tests.

The authors in [5] proposed a resistive touch screen technique to determine the position of a ball. This successfully eliminated the illumination effect that can cause an error in camera-dependent control systems. For the multivariable and complicated control system of a BPS, a touch screen and a rotating pneumatic cylinder are chosen in this paper instead of a camera and a stepper motor. The simulation results show that the system with the proposed control method has good dynamic and static characteristics. Not only has the fuzzy technique become a popular choice for the BPS, but there are also works that use a genetic algorithm with a neural network or a sliding mode controller to solve this nonlinear problem, as shown in [6]. In this paper, a genetic algorithm (GA)-based PIDNN controller (PIDNN) is proposed for the BPS. GA is used as a training weighting factor for a multilayer neural network, overcoming the disadvantage of the backpropagated algorithm (BP), which easily falls into partial extremes, and at the same time the advantage of the PIDNN controller, which has a simple structure and good dynamic and static performance.

Furthermore, the authors Y. Pattanapong and C. Deelertpaiboon in [7] propose a position control technique for the BPS using fuzzy logic with adaptive integral control. The aim is that the adaptive integral gain automatically adjusts its value and becomes active only when the position of the ball is within the specified distance error. This novel system takes advantage of the integral gain’s ability to eliminate steady-state errors and uses the fuzzy logic technique because it is simple without finding a mathematical-ematic model for this nonlinear system [8]. The current position of the ball is determined using a webcam mounted directly above the plate. Fuzzy controllers as advanced solutions are also described in [9,10]. In articles [11,12], the authors propose sliding mode techniques (adaptive back stepping control) with the strategy of fuzzy monitoring. They have experimentally found that adaptive back stepping control is more effective than conventional SMC control because it takes much time to achieve favorable tracking accuracy. In addition, one paper presents the use of FCMAC controllers [13] and feedback linearization controllers [14]. Another paper deals with disturbance modeling and state estimation for offset-free predictive control with state-space models [15].

In another paper, a virtual and remote laboratory for the ball and plate system is presented [16]. The authors in [17] proposed a control algorithm based on cascade PID and compared it with another control method. The paper shows the results of the accuracy of the ball stabilization and the influence of the philter used on the waveform. The application used to detect the ball position measured by the digital camera was developed using a cross-platform Net wrapper for the OpenCV image processing library—EmguCV. The aim of the paper [18] is to teach students the theory of control systems in an engineering context, through the design and implementation of a simple and low-cost ball and plate system. Students will be able to apply mathematical and computer modeling tools, control system design and the implementation of real-time software and hardware while solving a position control problem.

Numerous MPC algorithms have been used in the past for various industrial process controls, but also for numerous other processes. Examples of applications are: heating, ventilation and air conditioning systems [19], robotic manipulators [20], electromagnetic mills [21], servo motors [22], quadrotors [23], autonomous vehicles [24], modular multirotors, improved design of unmanned aerial vehicles [25,26].

A fast state-space MPC algorithm was presented in papers [27,28]. The paper [27] shows the development and modeling of a laboratory ball on plate process that uses the touchpad as feedback; a simplified process model based on a state-space process description. In paper [28], a fast state-space MPC algorithm is discussed. According to the authors, its main advantage is the simplicity of the computation: the manipulated variables are found online using explicit formulae, with the parameters computed offline; no real-time optimization is required. The articles [29,30] describe MPC algorithms with state-space process modeling and state estimation methods for these algorithms.

A practical approach is described in [31], but only for processes described by simple step-response models and by discrete transfer functions (i.e., difference equations). This work follows the idea presented for state-space models. Some specialized methods were developed to handle constraints in online MPC optimization that make it possible to use sampling times of the order of milliseconds [32].

A more advanced approach according to Lyapunov functions is discussed in the next papers. In both theory and practice, Lyapunov functions are an important tool for analyzing the stability of dynamical systems [33]. They guarantee the stability of equilibria or more generic invariant sets, as well as their basin of attraction. Numerous computational building approaches were created within the Engineering, Informatics, and Mathematics communities due to their usefulness in stability analysis. They apply methods such as series expansion, linear programming, linear matrix inequalities, collocation methods, algebraic methods, set-theoretic methods, and many others to various types of systems, such as ordinary differential equations, switched systems, non-smooth systems, discrete-time systems, and so on [34,35]. A method based on semi-definite programming is proposed in work [36] to estimate an invariance kernel with a target as large as possible by iteratively searching for Lyapunov-like functions. Central to the paper framework in [37] are Lyapunov invariants. These are properly constructed functions of the program variables, and satisfy certain properties—analogous to those of Lyapunov functions—along the execution trace.

Finally, the book [38] describes PID management of nonlinear systems based on passivity for the general engineering population towards the user-friendly approach. The E-book offers the material with minimal mathematical background, making it relevant to a wide audience. Familiarity with the theoretical tools reported in the control systems literature is not necessary to understand the concepts contained within. The latter was an inspiration to the authors of this research in order to adapt the topic of PID control to undergraduate study programs.

This paper describes the stages of designing and building a mechatronic BPS system with computer vision as feedback for educational purposes in STEM engineer education at the Karlovac University of Applied Sciences. The concept design of the depicted prototype emphasizes the avoidance of complicated mathematical methods and formulas in the ball control process. Aiming to achieve low-cost, well-documented, simple and easy implementation and good control precision, this paper proposes computer vision as feedback, via a Python OpenCV script PID controller with adjustable PID parameters to balance different positioning of given ball setPoints, as explained in the examples in the reference [39]. General knowledge of the theory of control of dynamical and nonlinear systems was used from the reference literature [40,41,42].

This paper’s contribution is divided into numerous thematic sections:The BPS mechatronic prototype’s original design was based on computer modelling capabilities for the manufacture of all robotic and auxiliary parts.To prevent machining metal pieces, all intended parts were produced utilizing 3D printing technology and prefabricated aluminium square tubes.Instead of elaborate mathematical models and settings for a nonlinear system, the Python OpenCV script with ready-made functions was used.A control technique is presented and implemented in the program code in accordance with the simplification of parameter manipulation by introducing ready-made Python script functions.A new interactive pop-up window for manipulating sensor outputs for process control, changing the colour, and setting the setPoint.

The following is a breakdown of the article’s structure.

Section 2 explains the methodology used in this research study. The computer design methods and procedures for building a laboratory BPS prototype are briefly described in Section 3. Individual robotic parts are designed in this area, including servo motor shaft holders, levers, and plate joints, with as few parts as possible. In Section 4, the Python script technique is detailed, with an emphasis on the ready-made functions for generating feedback by transforming a picture from a USB camera into a collection of ball position correction request data. The pop-up window software implementation in connection to the HSV standard color palette settings and the PID controller coefficients settings are discussed in Section 5. The findings of tests comparing the influence of the controller coefficients, the roughness of the substrate, and the amount of light are briefly presented in Section 6. Finally, Section 7 brings the article’s issues to a close.

## 2. Methodology

In this part of the paper, the authors discuss the methods they used in the research study. The chapter on methodology explains what they did and how they did it so that readers can assess the reliability and validity of the research. It covers the type of research conducted, how the data were collected and how the data samples were analyzed. It discusses which sensors and materials were used in the study and the reasons for choosing these methods.

The research design generally focuses on applied research with the aim of developing design techniques, building prototypes and implementing the control procedures. The authors wanted to increase the scientific understanding and solve the practical problem of controlling nonlinear systems more easily. In general, applied deductive research aims to test theory. However, in the case of this *case study* research, the focus is on demonstrating a new and simpler method for controlling a nonlinear system based on research and prototype implementation.

In collecting original data and analyzing the data, quantitative research was carried out with numerical results, while qualitative research is concerned with the descriptions and meanings of the experiments carried out. Both analyses were applied in this work. Quantitative research is expressed in numbers and diagrams, while qualitative research is expressed in words. It was used to understand design concepts, simple solutions for robotic servo arm design with dry bearing, observed uncertainties and inadequacies in the control system, and interpretation of the results of the numerous experiments. This type of research allows the reader to gain deeper insight into certain segments that may be misunderstood. Part of the qualitative method includes interviews with open-ended questions, observations described in words, and the literature reviews that explore similar concepts and theories of nonlinear systems control.

Of course, reliability and validity are usually terms used to assess the quality of research. The extent to which results are reproducible when the study is repeated under the same conditions cannot be guaranteed by the authors. The authors are aware that a reliable measurement is not always valid: the results may be reproducible but are not necessarily accurate.

An effective measurement was produced after determining a criterion variable. The correlations between measurement outcomes and criterion measurement results were not calculated expressly to test the criteria’s validity.

## 3. BPS Computer Design and Fabrication

The steps of the original BPS design and production phase are discussed in this section of the article. The BPS concept that was evaluated, designed, and chosen for production is essentially a clone of similar BPS solutions stated in the works [1,3,7,11,16,17,18], but with details similar to [11,27]. However, the “driving board” for the two servo motors had to be picked first. The well-documented Arduino UNO microcontroller board with two matched step actuators [43,44] was the obvious choice. The Arduino Uno is a low-cost, well-documented platform that was demonstrated to work in a variety of multi-platform applications. SolidWorks is well known as a software solution for computer-aided design (CAD) and computer-aided engineering (CAE) that is widely used in all cases of technical and engineering design [45]. Ultimaker Cura is the most popular printing software in the world [46].

### 3.1. Fabrication and Mounting

Because of its simplicity, the BPS prototype, shown in Figure 1, is made up of a dozen printed parts, including the servo motor first *plug*-arm, as shown in Figure 2 and Figure 3, servo motor *knee*-arm shown in Figure 4 and Figure 5, servo motor housing, shown in Figure 6 and Figure 7, the BPS bottom plate (five millimetres thick and upper plate of smooth Plexiglas two millimetres thick) shown in Figure 8 and Figure 9, camera housing shown in Figure 10a, tube slippers shown in Figure 10b, the base plate shown in Figure 11a, the central pillar of the BPS plate shown in Figure 11b, tube knees shown in Figure 12, and Arduino board base plate and mounting screws. The DC servo motor’s first robotic arm is designed and built with a central elliptical hole for the servo motor axle holder and a smaller round hole for the arm bearing shaft as shown in Figure 2 and Figure 3. This connection must take the entire servo motor axle holder, as well as the arm bearing shaft, without any air clearance.

The knee-arm shown in Figure 4 and Figure 5 is the second portion of the DC servo motor robotic arm, and it is built and parameterized to match the actual size of the BPS plate for the same horizontal distances from the plate’s centre, providing equivalent angular transmission from the DC servo motors [44]. The DC servo motor is held in place by the servo motor housing, shown in Figure 6 and Figure 7, which is screwed to the base plate shown in Figure 11a. The integrated tiny metallic ball in the top of the centre pillar of the BPS plate shown in Figure 11b provides a robust but flexible connection and ensures the BPS plate’s central location, as shown in Figure 11b. Furthermore, both servo motor knee-arms have small integrated metallic balls on top and support the BPS plate in a horizontal position as shown in Figure 11b by securely embracing the magnetic cups from the bottom of the BPS plate in a vertical position. A detailed description of the robotic system is available in [47].

The design steps of some BPS parts are displayed in SolidWorks software as final files for the Ultimaker Cura printing software in the following photographs from Figure 2, Figure 3, Figure 4, Figure 5, Figure 6, Figure 7, Figure 8 and Figure 9. Figure 2a,b illustrate the first part of the robotic servo arm, as an adjunct to the DC servo motor half-shaft, whose goal is a strong connection to the original output of the DC servo motor shaft on one side and a spaceless junction of the shaft with the jaws of a knee joint on the other side.

Figure 3 represents the first robotic servo arm “slice phase” in the printing software and the finished part of the servo arm after the printing process.

The crankshaft with the jaws of the second robotic arm of the servo motor is connected to the first servo handle by inserting the shaft into a small hole through both parts, as shown in Figure 4. The hole at the left side is a holder for a ball dry bearing.

The crankshaft “slice phase” in the printing software and the finished part with an installed magnet after the printing process are shown in Figure 5.

The Tower Pro MG995 DC servo motor housing design phases are shown in Figure 6.

The DC servo motor housing “slice phase” and finished part with built-in servo motor are shown in Figure 7a,b.

The BPS plate housing design phases are shown in Figure 8a,b.

The bottom BPS plate “slice phase” and the finished part with installed magnetic cups are shown in Figure 9.

The system sensor-HD USB camera is built into the white housing as shown in Figure 10a. The tube slippers for the two vertical square tube pillars are visible in Figure 10b.

The base plate assembly for the servo motors and central BPS pillar is visible in Figure 11a and three metallic balls for three magnetic cups under the BPS plate are shown in Figure 11b.

Figure 12 shows the elbows for the horizontal and vertical mounting tubes for the camera holder.

### 3.2. General BPS Design

This section of the paper describes the implementation of computer vision in the mechatronic education BPS prototype. During the project’s execution, which included the preparation of the student’s practical diploma thesis and subsequent experimentation by the co-authors in this paper, some limitations and flaws in the prototype, 3D print material and method, as well as difficulties in achieving stability when placing the ball in the desired position, were discovered. The purpose of this paper and project is to provide a basic and accessible experimental setup for learning, programming, and comprehending feedback control concerns in a real-case manual setting.

The mechatronic system described in the paper was originally designed, developed and programmed with the help of the student Tomislav Tropčić at the Karlovac University of Applied Sciences [40]. The sideways view of the experimental platform is shown in Figure 13 (top left and right). The system uses a USB HD camera as a feedback sensor, placed 160 mm above the controlled platform embedded in the camera holder, as shown in Figure 13. The 1920 × 1080 pixel (Full HD) camera captures 30 frames per second. Other technical data of the camera are: High-Speed 120 fps PCB USB2.0 Webcam Board 2 Mega Pixels, 1080P, OV2710 CMOS, Camera Module with 2.1 mm Lens, ELP-USBFHD01M-L21.

Three balls with identical sizes but different colours were chosen for the experiment, as indicated in Figure 13 bottom and left segments. Table tennis balls with a diameter of 40 mm were chosen in the following order: black, red, and orange. A smaller red ball with a diameter of 20 mm was utilized as a comparison, composed of a silicone mixture with a substantially higher mass. The ball was moved using a variety of materials with varying friction properties: 3D print material, two millimetre Plexiglas cover plate, white paper, and light grey sandpaper (180 particles per inch). The chosen materials had varying roughness values, which resulted in unequal resistance during the movement of the test balls over time. The white 3D printed platform plate is 150 × 150 mm and is supported by three supports, or pillars, the middle of which is vertically immobile and located in the geometric centre of the platform’s square surface. A simple dry “magnetic” bearing with a metallic ball and a magnetic cup on the underside of the platform in the geometric centre was designed to tilt the platform in both horizontal axes. When the DC servo motor’s two vertical robotic arms are raised and lowered, the platform tilts in firm contact with the robotic handle through a dry bearing on one side or the other. Servo motors are connected to the lower half of the motor with steerable arms with a wedge in the elbow, as shown in Figure 13 above and left. They are at a 90-degree angle to each other geometrically, and the grips are equidistant from the central fixed bearing. The servo motor handle’s horizontal portion (first arm) is attached to the servo motor protrusion, while the vertical portion (second arm with jaws) includes a spherical metallic ball glued to the top and a magnetic cup. Because the cup is fastened in the lower half of the steerable base, they form a firm and dry bearing that facilitates rotation.

A simple robotic lever system was created using a solid elbow and a shaft with a wedge diameter of 4mm as a dry bearing, in which both DC servo motors with a rotating angle of ±15 degrees transmit the same angular motion to the BPS platform.

## 4. Python Script

### 4.1. Computer Vision Issues

Performance in applications of recognizing patterns, forms, colours, and positions of objects is one of the most critical difficulties in the application of computer vision. Given the limited quantity of data available in robotics, the issues of choosing the right substrate, lighting, and methods for evaluating image and video quality without a reference are significant. Although simulations and visualization are crucial components in the preliminary phase of the scientific setup of an experiment, the algorithms utilized concern real applications rather than the development of mere theory.

Image formation, CCD camera resolution, advanced image features, real-time sampling frequency, binary vision, optical flow, image filters, object creation, epipolar geometry reconstruction, motion tracking, segmentation, grouping, and also recognition of objects are all unavoidable topics in computational vision in mechatronics. Advanced research in this scientific subject is enabled by the capabilities of software modelling of image processing techniques and approaches for object localization and geometric measurements. If the experimental setup is conventional, such as a USB HD camera, software for analysing and developing image processing functionality is becoming a powerful tool.

### 4.2. Image Converting Technicques

The description of the ready-made functions used in the Python script related to image converting techniques is given in the order in which the image obtained using the USB camera is processed. In order to get more images per second, in the program code, the resolution is halved to 640 × 480 pixels, so the number of captured images can be doubled, from 30 to 60 images per second. Ready-made Python image resolution function is defined as: “self.cam_width = 640, self.cam_height = 480”. The camera uses a USB connector to power and communicate with the computer.

### 4.3. List of Python Ready-Made Functions

VideoCapture object—VideoCapture()

When launching the application, it is necessary to create an object that will capture a video recorded with a USB camera. The application does not process the stored video (e.g., on the hard disk or memory card) but the stream of data that the camera records in real-time (live stream), to download a series of images from the camera (30 images per second), the so-called VideoCapture object. The VideoCapture object only needs to specify the camera number (0 = built-in, 1 = external USB camera) where the recording comes from. Algorithm 1 shows a fragment of the code. All other processing (reception, processing and image formation) will be performed autonomously “under the hood” of the ready-made function and thus free the programmer from a big job.
**Algorithm 1.** Fragment of the Python code: function cv2.VideoCapture(1).def start_cam(self):self.capture = cv2.VideoCapture(1)   # 0 for webcam, 1 for extern 

In this part of the code, it is necessary to define the dimensions of the images captured by the camera, and it is defined that the image is 480 pixels high and 640 pixels wide.

Colour model conversion from RGB to HSV—cvtColor()

All colours are obtained by using and combining colours in the colour palette. If we use the RGB (R-Red, G-Green, B-Blue) palette then we have three basic colours: red, green and blue. If each colour is written in 256 shades, then by a combination of available shades we get a palette of 16.7 million colours. Another colour representation (or colour space) is HSV (H-Hue or Tone, S-Saturation, V-Value or Brightness). RGB color space does not separate color and brightness information so brightness variations affect RGB channel values. HSV color space abstracts color from saturation and brightness and is suitable for color-based image segmentation [48]. The switching was carried out in secret because it is easier to get a binary image of the object when it is written in HSV format. The function is shown in the code fragment in Algorithm 2.

Noise image removal—GaussianBlur()

The next step is the process of removing noise from each image. The first step is blurring the edges of the image (Blur), using the Gaussian Blur function (blurring is performed using the Gaussian formula). When applied in two dimensions, this formula produces a surface whose contours are concentric circles with a Gaussian distribution from the center point. OpenCV documentation related to the Gaussian Blur states that the kernel size should be a positive and odd value. Higher values imply a more blurred image and vice versa. The authors decided to use a Gaussian kernel size of 11 × 11 pixels which is used by the OpenCV 2D filter function as the minimum size in order to convolve an image with the Discrete Fourier Transform-based algorithm [49]. The function is shown in Algorithm 2.

Binary image formation—inRange()

The captured image has a certain resolution (640 × 480 pixels), is converted to an HSV colour model and noise is removed. It is necessary to translate the image from a coloured to a black and white image without shades—where the pixel in the image is coloured with either black or white. It is necessary to determine which HSV formatted colours are converted to black and which to white. The utilized object tracking methodology detects the object based on the range of pixel color values in the HSV color space. The selected color will be displayed as white, while all other colors will be displayed as black in the binary image, as shown in Figure 15. The function is also shown in Algorithm 2.

Binary image noise reduction—erode()

The resulting binary image may have certain noises that are usually located at the boundary of the contour of the object (in the binary image). Applying the erode() function of the application will remove certain noise, but the consequence may be a reduction in the contour of the object; shown in Algorithm 2.

Contour thickening—dilate()

In order to amplify the contour of the object on the binary image, it is necessary to use the dilate() function, which will “thicken” the contour by a certain amount of pixels in order to be more clearly visible (Algorithm 2).
**Algorithm 2.** Fragment of the Python code: other conversion function with associated parameters.def process_frame(self, frame):self.hsv = cv2.cvtColor(frame, cv2.COLOR_BGR2HSV)# transform into HSV color space⋯  blurred = cv2.GaussianBlur(self.hsv, (11,11), 0) mask = cv2.inRange(blurred, self.color_low, self.color_high) # thresholdmask = cv2.erode(mask, None, iterations = 2) mask = cv2.dilate(mask, None, iterations = 2)

Object localization on a binary image—findContours()

After forming the binary image and the object, it is necessary to determine the contours of the object located in the image. The contours are passed to the application as a list of coordinates of the outer points that close the contour. There may be multiple contours in the image (intentionally, by mistake, or so) and then the application will look for the contour that occupies the largest area. The function is shown in code fragment in Algorithm 3.
**Algorithm 3.** Fragment of the Python code: function findContours().# find countours in the maskctns = cv2.findContours(mask.copy(), cv2.RETR_EXTERNAL# find the largest contour in the mask, then use # it to compute the minimum enclosing circc = max(cnts, key = cv2.contourArea)((x, y), radius) = cv2.minEnclosingCircle(c)

Minimal circle within the contour—minEnclosingCircle()

After locating the contour of the object, the smallest circle is entered inside it so that the coordinates of the centre and the size of the radius of the object can be determined. In this way, the centre and edge of the contour on the binary image are determined (Algorithm 3). The procedure requires that the radius of the contour be a minimum of 10 pixels in length, and after finding the contour, the application displays a circle and its centre so that the application user has an idea of where the application has located the centre of mass or geometric centre of the sphere. After determination, it is necessary to send the coordinates of the centre of the contour and the radius according to the function that controls the PID controller function: self.PID(self.setPointX), as shown in Algorithm 4.
**Algorithm 4.** Fragment of the Python code: drawing setPoint on screen cv2.circle() and self.PID().# only proceed if the radius meets a minimum sizeif radius > 10:⋯# length of min 10 pixel# draw setpoint on screen—5 pixel red dotcv2.circle(frame, (int(self.setPointX), int(self.setPointY)), 5, (0, 0, 255), −1)self.PID(self.setPointX, self.setPointY, x, y)# PID setpoint actual position in x, y,

All used ready-made functions: VideoCapture(), cvtColor(), GaussianBlur(), inRange(), erodes(), dilates(), findContours(), minEnclosingCircle) and self.PID() in parentheses can receive certain parameter values. Each function does a lot of work (calculations) and significantly simplifies the application and its use. For this reason, the number of lines in the program and consequently the size of the control program is significantly reduced.

After running the script all functions and parameters are prepared to locate and calculate the ball shape and find its geometrical center as the start setPoint (inputX, inputY). The Python script starts the motors and aligns the axes of the platform at the appropriate angles to align the stability with the initial start-position of the ball.

### 4.4. Python Control Script Design

Python’s control algorithm requires knowledge of past values. Proportional-integral control, for example, monitors the cumulative sum of differences between a setPoint and a process variable. Because the Python function disappears completely after feedback, the value of the cumulative sum must be stored elsewhere in the code. The problem with coding is figuring out how and where to store this information between call algorithms. For coding reasons, an object generator was created where certain parameter values can be received in parentheses. There are several ways to get value from a so-called number generator. One way is to use the next() Python function which executes the generator until the next yield expression is encountered and then returns the value.

Python script captures a series of camera images at 30 frames per second, approximately every 33 ms, which is the sampling rate of the ball position or the speed of calculating the position correction. Thus, the parameter dT is a time constant that correlates with the image processing speed, which is a PID controller iteration parameter. Algorithm 5 shows the code fragment where the time variable was defined.
**Algorithm 5.** Fragment of the Python code: time variable dT definition.# how long since we last calculated (dT definition)now = time.time()  # now = begining of application# change in time (dT = )dT = now − self.last_time⋯# print (dT)# save for next iterationself.last_time = now

### 4.5. Advanced Block Diagram of PID Controller

During the experiment and the selection of the most suitable colour, shape and size of the ball, as well as the surface of the plate, it was realized that the block diagram is not as simple as it seemed at first glance. Significant and unavoidable disturbances were observed, i.e., external influences that prevented the stable operation of the mechatronic system and the placement of the ball at a given setPoint. Interference functions were observed that cannot be accurately described mathematically but have proven to be influential because methods of reducing problems and attempts to eliminate them have led to better results and greater stability. For this reason, and shown in Figure 14, an improved block diagram control loop was proposed that highlights the locations in the CLC loop and the type of dysfunction or detrimental effect on ball position stabilization. First of all, a dysfunction (accidental disorder) is defined, which is denoted as *d*_1_(*t*), which represents mechanical imperfections and clearances of the handles that contribute to the increase in error.

Furthermore, another dysfunction *d*_2_(*t*) describes a group of functions within the software that, if inconsistent or unable to perform their task properly, increase position vagueness and introduce uncertainty and directly lead to significant problems and instabilities during position control.

The third influential quantity that contributes the most to the results of the experiment is the amount of scattering or light intensity. The system was shown to have the greatest stability if the illumination is adequate and light is scattered on the substrate from several sources and the original beam of the lamp is shaded. Each shadow of the ball from the light source significantly changes the colour shade of the ball on the edge of the ball and changes the contour image, which contributes to poorer recognition of the contour shape and consequently the creation of a binary image. It was observed that with a single light source although the system has a dispersive structure, the controller cannot stabilize the ball at all due to the above errors or conversion imperfections. In block diagram view, dysfunction *d*_1_(*t*) has a direct impact on the process (plate position) and form “steady state error”. Similarly, dysfunction *d*_2_(*t*) as “internal” uncertainty creates a cumulative effect on the Python output dataset (inputX, inputY) before the setPoint calculation process (setPointX, setPointY) and thus forms a “light shadow error”.

### 4.6. CLC Error Value Calculation

The equations in Algorithms 6 and 7 put into the Python script make significant progress, highlighting the capacity to generate ball control utilizing program ready-made calculations through functions and handle ball control without real physical hardware (external controller). The CLC comparison process generates error values for both the X- and Y-axes, errorX and errorY, which are defined in the computer code by parameters, as shown in Algorithm 6. In Algorithm 6, the phrase “inputX” refers to the ball’s real beginning location in the plate along the *X*-axis, while the term “self.setPointX” refers to the new ball position setPoint.
**Algorithm 6.** Fragment of the Python code: calculation of the error values.# error variables from the comparison processerrorX = self.setPointX − inputXerrorY = self.setPointY − inputY# print(errorX, errorY)

In most cases, a PID control system comprises two independent classic PID controllers connected by a single loop. The first manipulates the PWM control signal of the first DC servo motor to control the ball’s X position. The second, as illustrated in Algorithm 7, uses the PWM control signal of the second DC servo motor to regulate the Y position. Assuming the board has two axes, uniformity and ideal perpendicularity, the PID controller used the identical coefficients for both axes. Equation (1), as described in [40,41], is the canonical mathematical form in general theory:(1)$$u\left(t\right)=kPe\left(t\right)+kI\int_{0}^{t}e\left(t^{\prime }\right)dt^{\prime }+kD\frac{de\left(t\right)}{dt}$$
where term *e*(*t*) in Equation (1) is the errorX value in the Python script shown in Algorithm 7, term *e*(*t*′)*dt*′ is term self.error.SumX and term *de*(*t*)/*dt* is dErrorX shown in Algorithm 7. The control signal voltage *u*(*t*) is represented in the program as the control signal for operating the *X*-axis DC servo motor and is denoted by “angle X” according to Equation (1). The *X*-axis control signal is thus a sum of three terms. The voltage control signal is labeled “angleY” in Algorithm 7, similar to a *Y*-axis DC servo motor.
**Algorithm 7.** Fragment of the Python code: calculation of both axis PID control signals.# angle variablesangleX = self.zero_x + (errorX * self.kP + dErrorX * self.kD + self.kI * self.errorSumX)angleY = self.zero_y + (errorY * self.kP + dErrorY * self.kD + self.kI * self.errorSumY)

The coefficients of the PID controller *kP*, *kD*, and *kI* stated in Equation (1) were chosen and placed into the program code as default values during the optimization process, as shown in Algorithm 8. They can be changed during the experiment in the 0.001 value stages of the control application pop-up window.
**Algorithm 8.** Fragment of the Python code: PID controller coefficients default values.# PID coefficients self.default_P = 0.033self.default_D = 0.023self.default_I = 0.001

Proportion gain, term *kP*, is responsible for the corrective reaction and is used to identify the difference between the desired and actual values, as shown in [40]. With the increasing gain, the error lowers as the system gets more oscillatory. To determine the integral value, the integral term *kI* is used to calculate all previous error values and then integrate them. Integral action can also be thought of as a way to automatically generate the bias term in a proportional controller [41]. When the error value is removed from the system, this integral term stops growing. Based on current values, the derivative *kD* is used to anticipate future expected error levels. If the system has a fast rate of change, which is also reliant on the derivative component, the controlling effect can be amplified. The entire value of the required correction is obtained by combining these three operations. The PID controller’s constants *kP*, *kI*, and *kD* can be adjusted both in the program code and in the graphical visualization space boxes shown in Figure 15. As shown in Algorithm 9, calculated control signals for Arduino board as PWM driving platform for both DC servo motors are presented below.
**Algorithm 9.** Fragment of the Python code: formatted values for the Arduino board.# send to Arduino board − X and Y control signals arduino.write((str(angleX) + “,” + str(angleY) + “\n”).encode())# print(angleX, angleY)

## 5. Dynamics and PID Control Issues Overview

There are numerous methods for controlling a dynamic system [40,41]. The philosophical principles that underpin these methodologies can be broadly classified into three types for the sake of this *case study*: descriptive, model-based, and myoptic. Descriptive techniques presume that a controller is provided, and the purpose is to determine whether the controlled system meets certain stability requirements. Simulating the system or running it under a variety of operational light and surface conditions and seeing the outcomes are examples of empirical tests. After the control parameter is chosen at the current moment, a myoptic approach will look at the direction of a ball movement in state space.

The core algorithm of 1D control systems, i.e., the *X*-axis control, is proportional-integral-derivative control [40]. It is the most studied class of controllers due to its simplicity, and it is almost always the first thing to test on a new system [41]. Despite the fact that it lacks a model and is short-sighted, it may operate admirably with a few manual tweaks.

During experiments, it was discovered that the ODE solution is a damped harmonic oscillator. This oscillatory behaviour means that the oscillation will overshoot the setPoint for any nonzero setPoint starting state. Furthermore, the frequency of oscillation *ω* is dependent on both the gain coefficients and the system coefficients. Lower *kI* values will minimize and finally eliminate oscillation, although recovery from steady-state error will be slower.

A comparable consideration of the PD control problem for a second-order system yields the damped harmonic oscillator system, which is also featured in the experiment. Because derivatives can be approximated using finite position differencing: *x* = *x*(*t*) − *x*(*t* − *dt*)*dt*, derivative estimation mistakes are an issue. The derivative contribution, however, is more sensitive to measurement noise than position estimations since *t* is tiny and in the denominator. As a result, the derivative term varies, leading the control to track less precisely and in an irregular manner.

### 5.1. BPS Visualization and Control

In this part of the paper, a discussion is focused on visualizing the position of the ball after activating the application and managing the position of the ball. First, during the experiment, it was proved that of the three selected balls, the highest quality conversion to a binary image and the entire image processing covers the case of the modified orange colour (HSV format parameters–0/77/115/51/253/255) with a slight deviation from the entered “default” value in relation to the value entered in program code (HSV–default 1/77/115/61/153/255). The red colour (HSV–default parameters 121/157/86/243/255/255) did not give sufficient response quality, despite parameter modification, which resulted in an increase in the value of the disturbance function *d*_2_(*t*) and ultimately too much error and deviation in the calculation, which manifested itself as the possibility of setting the red ball to a given default setPoint on the platform. The black ball, despite having the strongest colour contrast in its parameters (HSV–default 0/0/0/25/25/25), could not be recognized at all as a shape or contour in the HSV standard, probably due to poor lighting quality.

### 5.2. Application “Ball Tracking”

Figure 15 shows an interactive “Ball Tracking” pop-up window that serves as the controlling device window for the mechatronic BPS prototype. It is possible to control the process with different critical parameters using designed functions that are performed on the screen. The centre of mass estimated in Python script as the true centre of the orange ball is represented by the small white dot, which is five pixels wide. The normal and computed binary variants of ball pictures are displayed in the upper right corner of the ball tracking window, as shown in Figure 15a,b. Although the small white dot on the computer screen symbolizes the ball’s centre, clicking on a new place on the plate establishes the ball’s desired position as a small red dot, also five pixels wide, as seen in Figure 19. In a Python script, the equations for calculating PID error values automatically generate the correction value for both the X- and Y-axes, balancing the BPS plate with both actuators.

Six HSV palette sliders are located on the left side of the Ball Tracking pop-up window, allowing fine customization of colour hues for the best binary conversion. Figure 15a shows a real-time image from a USB camera with an orange ball that showed the best responsiveness and presentation of the image live stream during the experiment in the upper right corner. Below this section are three frames or “space boxes” for fine-tuning the controller PID coefficients in 0.001 unit increments or the “Reset PID” option for default values (stored in script). The object search (Start/Stop Tracking) is controlled by two square space boxes in the lower-left corner, while the right button controls the servo motors (Start/Stop Motors). Furthermore, at the very bottom of the interactive window, there is a very handy option to modify the horizontality of the plate manually in relation to the unevenness of the substrate on which the prototype is positioned. In the upper left corner, the image’s sampling frequency in milliseconds is also shown (insert value 32 in Figure 15a). A binary figure of the ideal shape depicts the identical position of the ball in Figure 15b. By pressing the “Show Thresh”/”Normal View” button, you can change the images.

As for the servo motor’s robotic arms, the calibrated mechanical “zero horizontal position” of the plate is a default angle of 37 degrees for both actuators, as shown in Figure 15a,b. If necessary, the “zero position” can be adjusted in one-degree increments within the “Calibrate” *X*- and *Y*-axis space boxes. Angle control is limited to ±15 degrees on both axes. In the experiment, the proportional coefficient *kP* is chosen at a value of 0.03, the coefficient *kD* is chosen at a value of 0.02, and *kI* is chosen at a value of 0.01 or less.

Two further pop-up screens were added to the Python script, which initiate the graphical representation, time period charts, and numerical matrix representation of the relevant parameters for future mathematical analyses. Figure 16 shows, for example, a 6-s time period chart with a graphical representation of the actual and selected position setPoint, as well as shaft angle value as PID control signal. For a better understanding of the dynamics and stability of the BPS system, the time period of the strip chart is extended to 20 s in Figures 17, 21 and 22.

The second manageable pop-up window in the Python code but on the background screen represents numerical data of parameters shown in the figures in the same time period for additional analyses, if needed, respectively.

## 6. Experimental Results

Following the creation of the prototype, it was required to functionally verify the work and optimize all of the Python script’s functionalities using actual BPS prototype components. After multiple revisions, the BPS system’s functional operation was achieved, allowing the setting of a ball setPoint anywhere on the plate’s surface (150 × 150 mm). The “sliding” of a smooth ball on a smooth Plexiglas plate was the first item that was noted. As indicated in the advanced block diagram in Figure 14, this is a verified flaw of the mechanical system according to the plate smooth surface, generally referred to as “steady-state error” or dysfunction *d*1(*t*).

In this research, a graphical depiction of the ball movement exclusively along the horizontal *X*-axis was used due to the easier explanation and highlighting of crucial elements connected to the control problem. There is a usual “exceeding” of the value of the ball position in both directions during the first experiments. Several items sparked suspicion: specific PID values, sliding on a flat surface, and mechanical clearances. The process of altering the X-position of the ball from one side of the plate to the other along the *X*-axis by roughly 250 pixels, as shown in Figure 16, is typical of the first series of tests. The graph in Figure 16 displays the ball’s actual beginning X-position: 100 pixels at 0 s (blue line) and the newly selected (setPoint) position at 390 pixels (red line) farther along the *X*-axis. Additionally, *kP* = 0.033, *kI* = 0.010, and *kD* = 0.023 are the controller constants. The PID control system can keep the ball within an overshoot of ±24 pixels, or around ±8 mm, using this set of constants. The controller does 32 control adjustments every second, and there are exactly 16 signal orders for the *X*-axis servo motor in each exceeding of the ball’s setPoint, as shown in the lower section of Figure 16. Angle variations are approximately ±4 degrees. Without a doubt, a typical example of unsteady system operation with the harmonic frequency of one Hz is illustrated [50].

In the following studies, despite varying PID parameters, no substantial stability was attained. Several mechanical flaws were discovered after the study. When using additive technologies, such as 3D printing, keep in mind that due to the thermoplastic material’s characteristics, deviations in all three axes might occur throughout the printing and cooling process. This is mostly dependent on the thickness of the filament material to be applied, with rises and depressions being observed in thin layers of big surfaces, such as tiles. In this scenario, mechanical levelling was required to mechanically polish the surface of the printed BPS bottom plate. Furthermore, a new two-millimetre-thick Plexiglas plate with a sandblasted surface was added instead of smooth Plexiglas. Figure 17 displays the BPS chart for a 20-s period with the X-setPoint changing from 110 to 400 pixels. Within the first two seconds of the average range of ±12 pixels, rough stabilization of the location is noticeable, followed by fine stabilization of the position after four to five seconds within the limits of ±6 pixels. The controller coefficients in this experiment are *kP* = 0.033, *kD* = 0.022, and *kI* = 0.001.

During the studies, it was discovered that lighting had a significant impact on the sensor system’s operation. As seen in Figure 18, strong light sources on one side of the ball were demonstrated to destabilize the BPS system (d). Figure 18 depicts multiple scenarios based on the ball’s strong and low illumination: (a) weak illumination conditions; (b) and (c) one dominant light source. The physical interruption of the ball’s border contour as a result of this effect leads to an inaccurate determination of the ball’s centre of mass in the binary representation of the image.

This makes it difficult to use the Python method “inRange ()” as previously described. As indicated in the advanced block diagram in Figure 14, it was important to describe and document these sensor feedback defects caused by a poor image conversion system, which is referred to as “light shadow error” or *d*_2_(*t*) dysfunction.

Several smaller discrete shaded light sources were added to mitigate this negative effect, and the control precision was greatly enhanced.

This increased the amount of light directed towards the ball, which at the time had no visible shadow. A resolution issue, known as “particular distance mistake”, was also discovered. It is the control system’s failure to recognize a new sphere centre position setPoint that is very close to the existing actual setPoint. This could be classified as a sort of hysteresis, i.e., sensor or computer vision recognition insensitivity.

The largest specified distance error was found to be 6-pixels or roughly two millimetres. This is the same as the diameter of the red dot that represents the sphere’s center of mass. The control instruction to move the ball 4 pixels in the horizontal *X*-axis direction is shown in Figure 19, however, there is no answer since the needed setPoint offset is within the defined distance error, which is the size of the sensor recognition error. Figure 19’s time graph on the right side displays the setPoint value of the ball at “201 pixels” on the panel and no actual signal from the controller. The blue line represents the signal noise from the ball’s actual position sensor, which has an average value of 197.3 pixels and a variance of ±0.2 pixels (±0.07 mm) and this is the proven sensitivity of the CCD sensor.

Figure 20 shows the residual specific distance error after the controller correction process: the setPoint position demand in the horizontal *X*-axis direction is 12 pixels (four millimetres). After two seconds of signal control, there is stabilization and residual dislocation of the real ball position of around two pixels, which is about 0.7 mm distance.

Many variants of the controller coefficient were developed in order to further improve the stability of the BPS system. The system works very quickly and nervously with large oscillations and the inability to stabilize the ball for a long period, roughly five seconds, when an integral coefficient *kI* is present, as illustrated before in Figure 21. The controlling process is greatly enhanced when the integral coefficient *kI* is excluded from the equation. The controller coefficients in this experiment, shown in Figure 22, are *kP* = 0.030, *kD* = 0.020, and *kI* = 0. The specific error of the final X-position distance is still occasionally seen in the steady location of the ball. The dislocation of the ball in a stable position 6 to 9 pixels (two to three millimetres) distance from the defined setPoint may be seen in the graph in Figure 22.

With practically every start of the ball position adjustment, the absolute angle correction of the DC servo motor with a maximum permissible correction of ±15 degrees can be seen in the lower graph in Figure 22. The proportional and derivative parts of the controller’s usual activity can be seen as characteristic control signals to the actuator in the lower graph in Figure 22.

Of course, in the absence of integration contribution, there is a delay in position control in the case of PD controllers, about 0.15 s after initiation, but the BPS system has considerably superior stability. According to iteration method frequencies, the frequency of the controller control signal directed to the *X*-axis DC angle correction actuator is 32 control actions per second, as shown in Figure 22.

## 7. Conclusions

The implementation of the BPS prototype as a laboratory platform for the education of STEM engineers is discussed in this study. In addition, the design and implementation of software and hardware are explored in detail. The computation time of an open-source control system based on Python scripts that permits the usage of ready-made functions from the library is quite short. Because of the OpenCV environment, the calculation may be made as simple as possible. The OpenCV technique was found to work when applied to the BPS process, however, it is important to improve the system in comparison to other publications in order to eliminate or partially minimize the influence of the amount of the disturbances indicated as mistakes in the improved block diagram. Because of the dynamic features of the mechatronic prototype and the circumstances surrounding suitable lighting, the PD algorithm proved to be more successful than the conventional PID solution. Because of the required amount of consistent light illumination, choosing an HD camera as a sensor for the control system feedback proved to be quite challenging.

A quantitative study was carried out with numerical results in collecting original data and analysing the data, whereas qualitative research was concerned with the descriptions and meanings of the tests carried out. In this study, both analyses were used. Qualitative research is expressed in words, whereas quantitative research is expressed in numbers and graphs. It was used to grasp design principles, simple solutions for robotic servo arm design with dry bearing, control system observed uncertainties and inadequacies, and interpretation of the results of multiple trials.

The scientific approach always seeks some categorical views and evidence and even doubts that open up opportunities for other research teams to investigate, confirm, or deny such phenomena more deeply. Additionally, the scientific approach always requires that readers from the presented paper can assess the reliability and validity of the research.

Nevertheless, the authors hope that the presented works will inspire readers and students to develop new methods and applications of machine vision and computer vision for industrial and non-industrial applications, as the authors will undoubtedly continue their research on the BPS mechatronic platform and control algorithms. The selection of various control algorithms and the usage of a resistive touchpad as a feedback sensor are the most likely directions for future study.

## Figures and Tables

**Figure 1 sensors-22-01875-f001:**
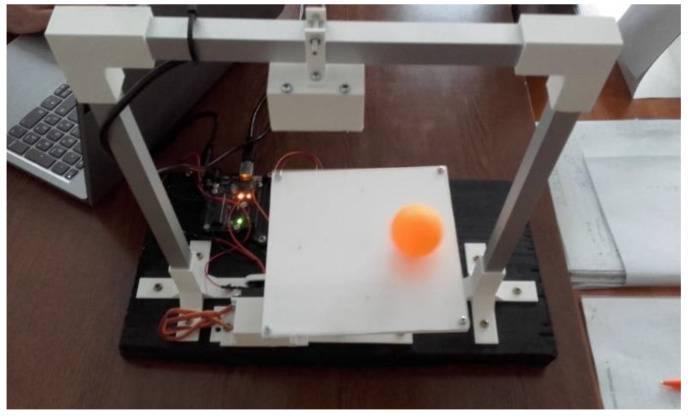
Laboratory educational BPS prototype in action.

**Figure 2 sensors-22-01875-f002:**
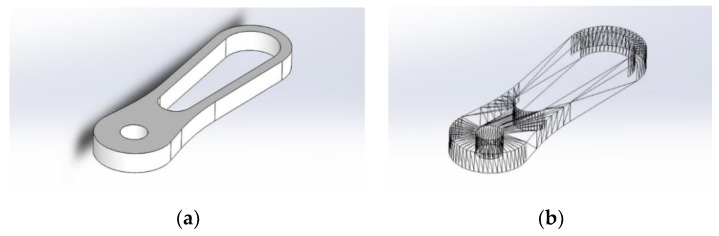
Servo motor first servo arm: (**a**) Visualization in SolidWorks; (**b**) SolidWorks mesh model.

**Figure 3 sensors-22-01875-f003:**
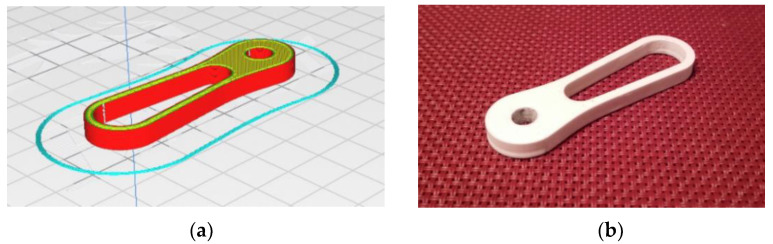
Servo motor first servo arm: (**a**) Visualization in Ultimaker Cura software; (**b**) Actual 3D print.

**Figure 4 sensors-22-01875-f004:**
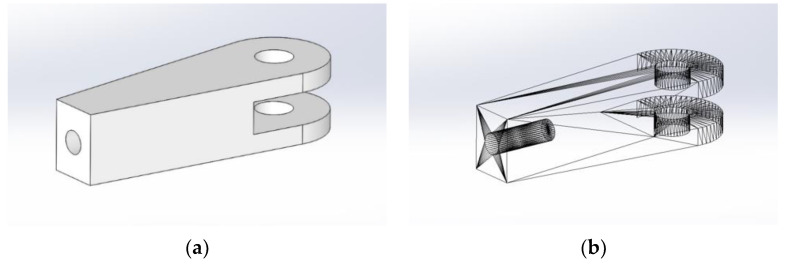
Servo motor crank knee-arm: (**a**) Visualization in SolidWorks; (**b**) SolidWorks mesh model.

**Figure 5 sensors-22-01875-f005:**
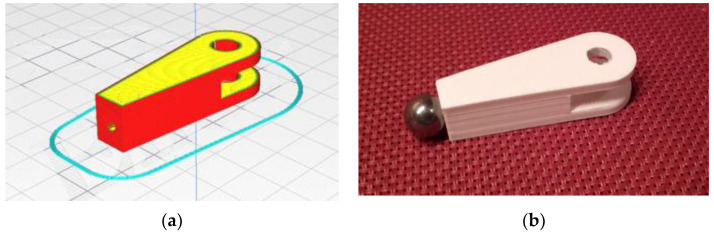
Servo motor arm crank part: (**a**) Visualization in Ultimaker Cura software; (**b**) Actual 3D print.

**Figure 6 sensors-22-01875-f006:**
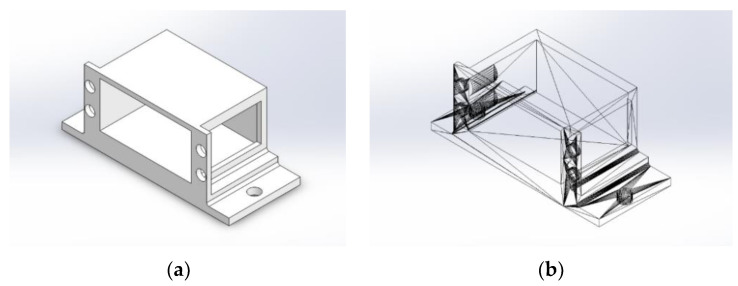
Servo motor housing: (**a**) Visualization in SolidWorks; (**b**) SolidWorks mesh model.

**Figure 7 sensors-22-01875-f007:**
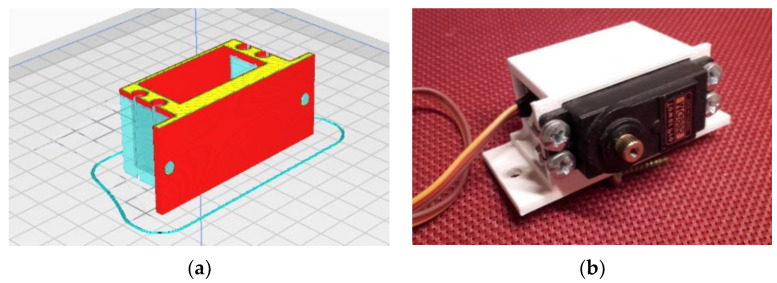
Servo motor housing: (**a**) Visualization in Ultimaker Cura software; (**b**) Actual 3D print.

**Figure 8 sensors-22-01875-f008:**
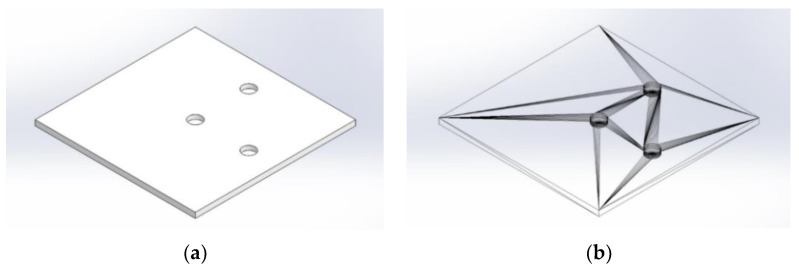
BPS plate bottom view: (**a**) Visualization in SolidWorks; (**b**) SolidWorks mesh model.

**Figure 9 sensors-22-01875-f009:**
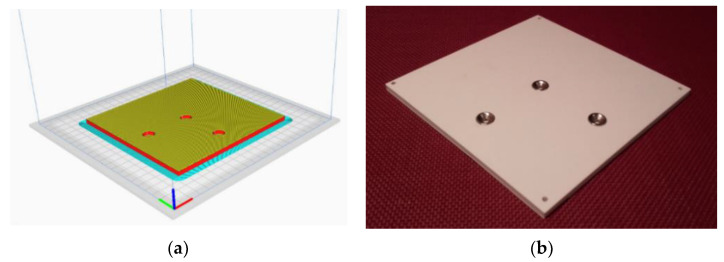
Bottom view of the BPS plate: (**a**) Visualization in Ultimaker Cura software; (**b**) Actual plate print with built-in magnetic cups for spaceless joint to the magnets.

**Figure 10 sensors-22-01875-f010:**
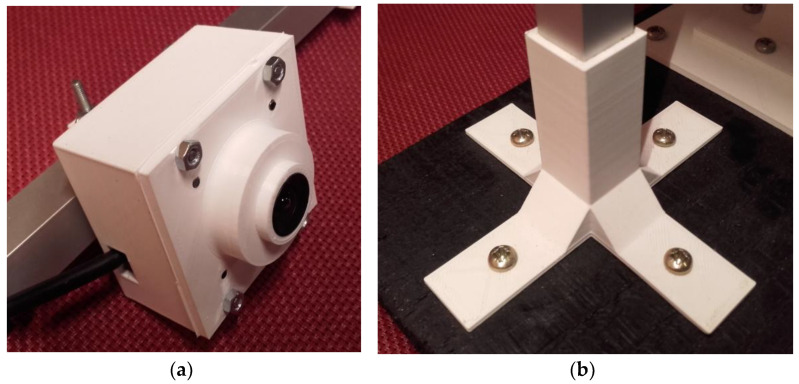
Printed BPS system parts: (**a**) USB camera housing; (**b**) USB camera tube slippers.

**Figure 11 sensors-22-01875-f011:**
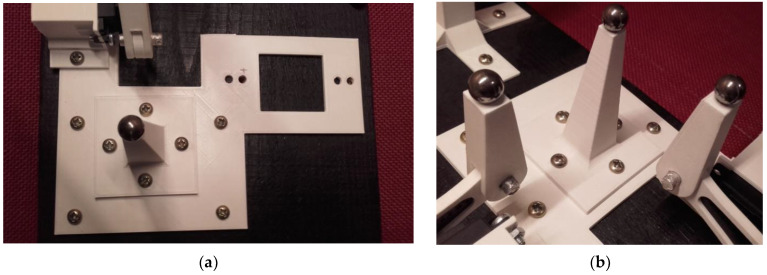
Printed BPS system parts: (**a**) Base plate; (**b**) Three pillars; central fixed pillar and two vertical servo motor knee-arms.

**Figure 12 sensors-22-01875-f012:**
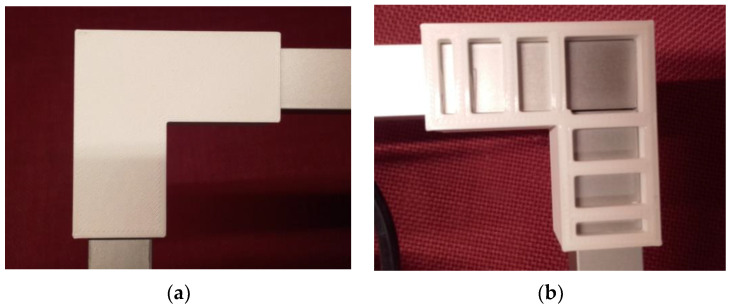
Printed square elbow for the camera holder tubes: (**a**) Front view; (**b**) Rear view.

**Figure 13 sensors-22-01875-f013:**
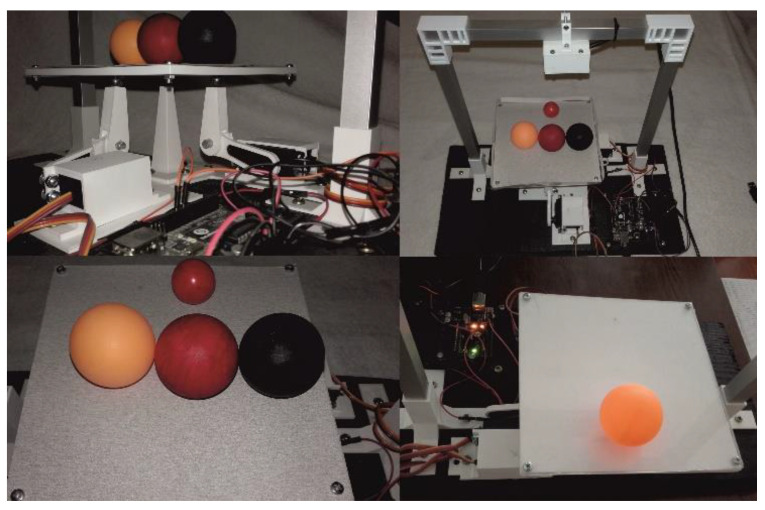
Educational BPS system with three experimental balls.

**Figure 14 sensors-22-01875-f014:**
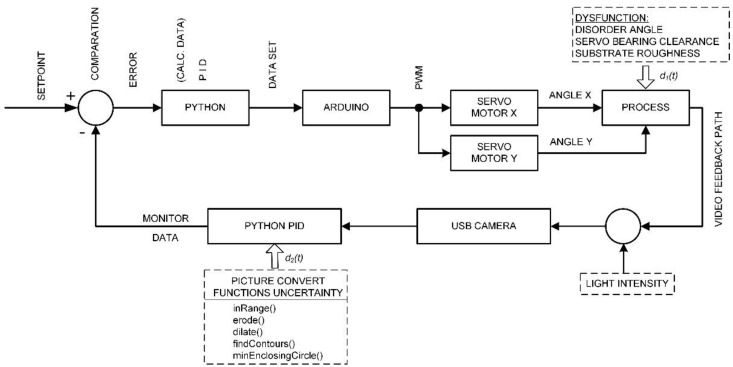
Advanced CLC control diagram scheme.

**Figure 15 sensors-22-01875-f015:**
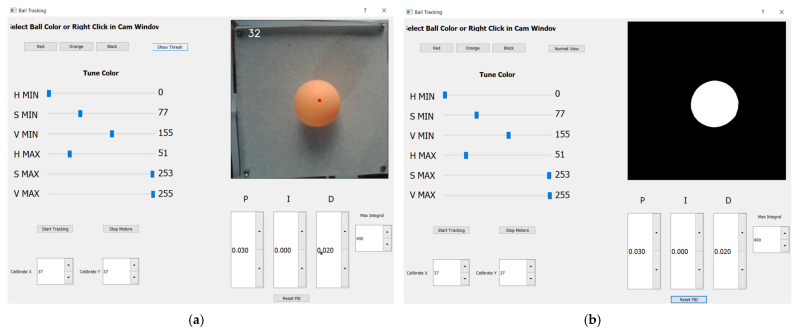
“Ball Tracking” pop-up window: (**a**) “Normal View” shows actual camera live stream image; (**b**) “Show Thresh” shows binary image (white ball contour on the black plate).

**Figure 16 sensors-22-01875-f016:**
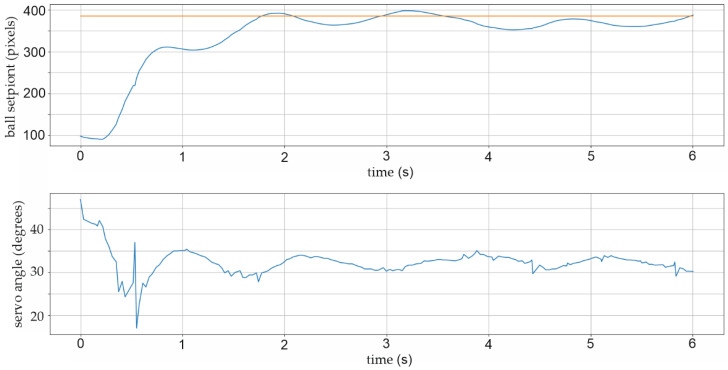
Experiment result: BPS printed bottom plate and upper Plexiglas smooth plate.

**Figure 17 sensors-22-01875-f017:**
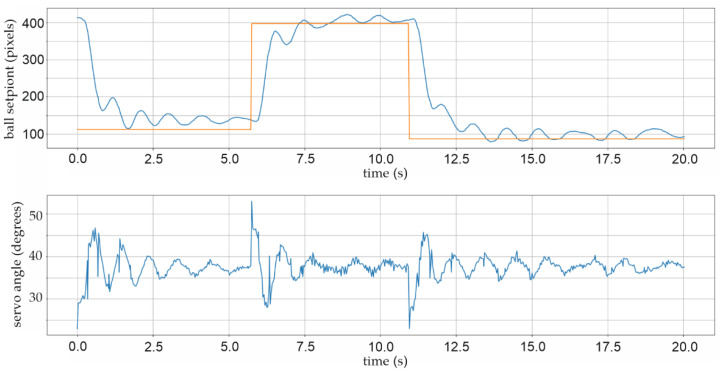
Experiment with the rough BPS plate surface.

**Figure 18 sensors-22-01875-f018:**
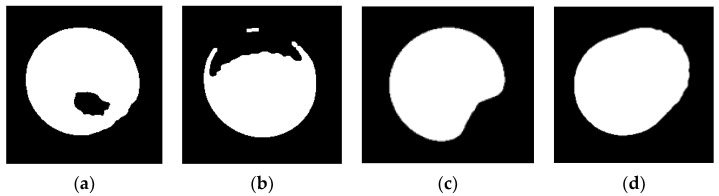
Examples of “light shadow error”.

**Figure 19 sensors-22-01875-f019:**
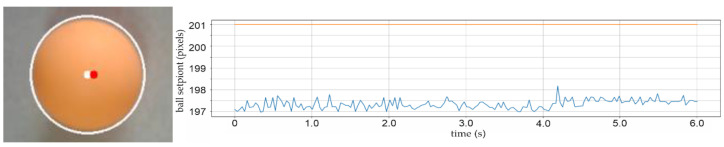
Specific difference between the distance of the actual (red dot) and setPoint (white dot) ball position.

**Figure 20 sensors-22-01875-f020:**
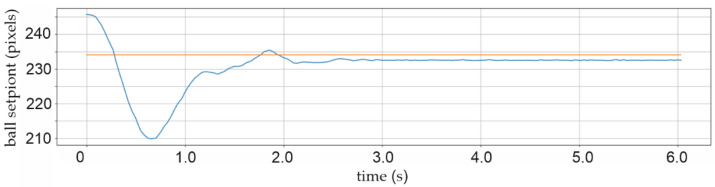
Experiment with residual ball position error-specific distance error.

**Figure 21 sensors-22-01875-f021:**
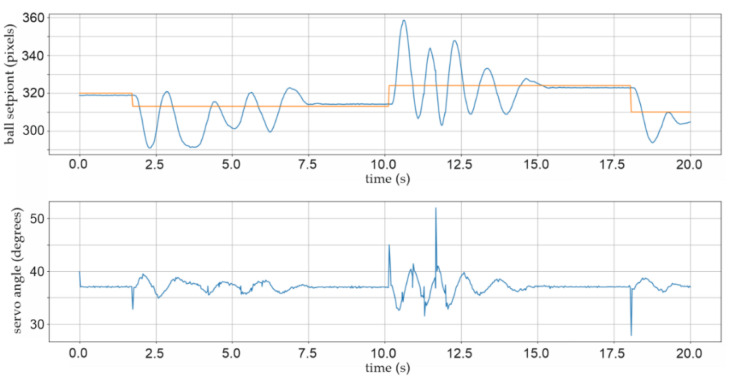
Experiment with *kP* = 0.030; *kI* = 0.010; *kD* = 0.020; ball stabilization after five seconds.

**Figure 22 sensors-22-01875-f022:**
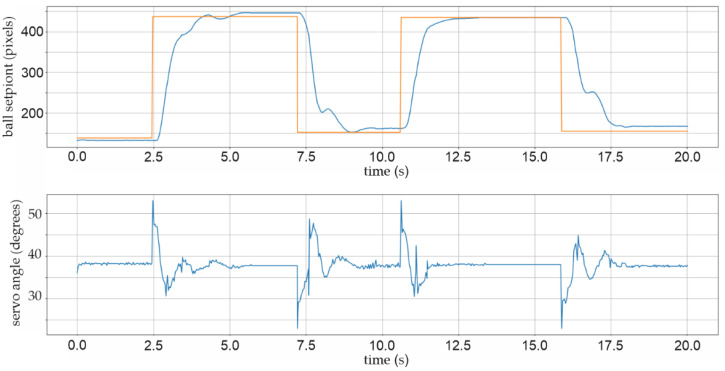
Experiment with *kP* = 0.030; *kI* = 0; *kD* = 0.020; ball stabilization after two seconds.

## Data Availability

The data presented in this study are available on request from the corresponding author.

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
