# Peer review of "Design and Implementation of a Ball-Plate Control System and Python Script for Educational Purposes in STEM Technologies"

_sensors, 2022, doi:10.3390/s22051875_

Round 1

Reviewer 1 Report

This paper is the description of a student project on building a Ball-Plate system for educational purposes. The project makes use of a 3D printer for the construction of various parts, of an Arduino UNO processor for the control of two servo-motors and of a web-camera for the purpose of following the ball in a PID control system. Ball detection is based on OpenCV functions.

The paper does not represent mature scientific work. It is merely an extensive report on a technical project. It lacks solid scientific contribution and it is only marginally related to sensors. The sensor part consists of a web-camera and its operation is trivial.

As a technical text, this report is suffering from unnecessary detail. Lengthy descriptions of well-known tools, like SolidWorks software, OpenCV library and its functions, or the Arduino UNO microprocessor are not suitable content for a Journal publication. This information can be found anywhere in the web and should only appear as a concise description of the methodology, without details. Similarly, fragments of code copied from a terminal screen should not appear as part of the publication. A journal publication should be fundamentally different than the presentation of a technical project in a web blog. We only present new ideas that contribute to the scientific endeavor in our field of interest.

The use of English in this report is rather below the expected standard for a journal publication. The paper should not only be fundamentally shortened to what really serves the presentation of its scientific contribution, but it should also be corrected by a fluent colleague, in terms of grammar and syntax. There is many a point where the poor use of language obscures the clear presentation of details.

The literature is rather weak, consisting mainly from conference proceedings or technical reports. However, one would expect the literature to be based on a solid body of journal papers, given the actual scientific problem which is the stability of a well-known automatic-control system.

My final comments are about the title of the paper and the abstract. The title could be merely “Design and implementation of a Ball-Plate control System for educational purposes”. As it is, the title is very generic and does not represent the actual content. I would also advise the abstract to be revised and shortened. It should contain the essence of the scientific contribution achieved in the work, instead of expanding on the challenges of Ball-plate and other control systems.

A concise and mature version of this work could be candidate for a conference or journal on educational tools. However, the educational value of the prototype should also be investigated and presented in a valid way.

Author Response

Thanks

Authors

Reviewer 2 Report

The manuscript describes an original ball-on-plate laboratory process. The process is intended to be used in engineering courses. In general, the manuscript is very interesting and quite well written. In my view, it may be accepted provided that the authors take into account the following issues:

  1. We do not know if the whole project is planned to be open-source, i.e. everybody will be able to copy the authors’ work. Please state it clearly (if it is true).
  2. In the case of ball-on-plate laboratory process with a camera used to determine the ball’s position, the delay introduced by the camera and image processing is an important problem. Unfortunately, the authors do not discuss this issue. Do you observe the delay? What is its influence on control quality?
  3. The authors use only the PID controller. The process is multivariable and two independent PID controllers do not give excellent control. It would be best to implement and present results for the LQR controller. If it is impossible, the software used should be extended for user-defined controllers.
  4. Please discuss all figures in the text. It is necessary to write: The Tower Pro MG995 DC servomotor housing design phases are shown in Figure 6. Please do not use the form: Tower Pro MG995 DC servomotor housing design phases (Figure 6). I only give one example, as the error occurs several times, please correct the whole text.
  5. Python scripts: please do not use a black background. Many people, including myself, do not see clearly dark letters on black background. Use white or light grey background.
  6. Figures 16, 18, 19 and 20: the font size is too small. Please use bigger fonts or please use smaller plots, that do not require a significant decreasing.

Author Response

Thanks

Authors

Round 2

Reviewer 1 Report

This paper is a report on a valid design and implementation problem with scarce actual scientific contribution on sensors. It is written as a good tutorial on how the system was designed and built and assesses its behavior. The system constitutes an example of good engineering and is well described and documented. It can be replicated by third parties and it can inspire enthousiasm on similar implementations.

Some of my advice has been taken, some is not, the authors can still refer to my original assessment. The paper can be further refined with respect to language and with respect to the rules of technical writing.

The most important thing is that the scientific contribution still is not clear. Please be careful that original design and 3D printing and using openCV do not constitute scientific contributions, unless they achieve original or outstanding behavior.

Author Response

Ad.1. The title of the article has been altered to reflect the original reviewer's opinion, as recommended.

Ad.2. Addition of 10 new scientific books and papers to the reference list.

Ad.3. The paper now has a new Section 2: Methodology.

Ad.4: Experiment’s description has been enhanced further.

Ad.5. Section 7- Conclusion has been modified and extended.

Ad.6. In an internet program, the paper's text is grammatically paraphrased into English; https://quillbot.com/ and https://www.grammarly.com/